# Selective photoelectrochemical oxidation of glycerol to high value-added dihydroxyacetone

Dong Liu [1], Jin-Cheng Liu [2], Weizheng Cai[1], Jun Ma[3], Hong Bin Yang[1,4], Hai Xiao [2], Jun Li [2], Yujie Xiong[3], Yanqiang Huang[5] & Bin Liu[1]

It is highly profitable to transform glycerol – the main by-product from biodiesel production to high value-added chemicals. In this work, we develop a photoelectrochemical system based on nanoporous $BiVO_4$ for selective oxidation of glycerol to 1,3-dihydroxyacetone – one of the most valuable derivatives of glycerol. Under AM 1.5G front illumination (100 mW cm$^{-2}$) in an acidic medium (pH = 2) without adscititious oxidant, the nanoporous $BiVO_4$ photoanode achieves a glycerol oxidation photocurrent density of 3.7 mA cm$^{-2}$ at a potential of 1.2 V versus RHE with 51% 1,3-dihydroxyacetone selectivity, equivalent to a production rate of 200 mmol of 1,3-dihydroxyacetone per m$^2$ of illumination area in one hour.

[1] School of Chemical and Biomedical Engineering, Nanyang Technological University, 62 Nanyang Drive, Singapore 637459, Singapore. [2] Department of Chemistry and Key Laboratory of Organic Optoelectronics & Molecular Engineering of Ministry of Education, Tsinghua University, Beijing 100084, China. [3] Hefei National Laboratory for Physical Sciences at the Microscale, Collaborative Innovation Center of Chemistry for Energy Materials (iChEM), School of Chemistry and Materials Science, University of Science and Technology of China, Hefei 230026, China. [4] Institute for Materials Science and Devices, Suzhou University of Science and Technology, Suzhou 215009, China. [5] State Key Laboratory of Catalysis, Dalian Institute of Chemical Physics, Chinese Academy of Sciences, Dalian 116023, China. Correspondence and requests for materials should be addressed to H.X. (email: haixiao@tsinghua.edu.cn) or to Y.X. (email: yjxiong@ustc.edu.cn) or to B.L. (email: liubin@ntu.edu.sg)

ncreasing global energy demand and environmental concern drive the development of fuels and building-block chemicals production from renewable feedstocks and green energy sources[1–6]. Biodiesel manufactured from vegetable oils and animal fats has become increasingly attractive because of its environmental benefits[7–10], which is now produced at million tonnes of oil equivalent per year. Production of 100 pounds of biodiesel brings out 10 pounds of glycerol as a by-product. With increase in global biodiesel consumption, efficient utilization of glycerol becomes an urgent and highly profitable market[11–17]. Glycerol has many oxidation derivatives, among which, 1,3-dihydroxyacetone (DHA) is most valuable. DHA is widely used in cosmetic, pharmaceutical, fine chemical, and food industries, which costs US $150 per kg while the price for crude and refined glycerol is only US $0.11 and US $0.66 per kg, respectively[18,19]. Commercial DHA is mainly produced via a microbial fermentation process, which is complex, time-consuming, and subject to substrate inhibition[15,20]. Besides, selective catalytic conversion of glycerol to DHA could be achieved by thermal oxidation or electrochemical oxidation on noble metal catalysts (e.g., Pt, Pd, Ag, and Au) with adscititious oxidants[21–28], which are not only cost-intensive, but also produce DHA at very low production rates with poor DHA selectivities.

Photoelectrochemical oxidation provides another promising approach, which combines both renewable feedstock and green energy source together to make building-block chemical and clean fuel at the same time. In this study, we demonstrate that photoelectrochemical oxidation of glycerol to DHA could be achieved with high selectivity at ambient temperature and pressure on porous $BiVO_4$. At pH = 2, under AM 1.5G front illumination (100 mW cm$^{-2}$), the porous $BiVO_4$ photoanode is able to produce about 200 mmol of DHA per m$^2$ of illumination area in 1 h at a potential of 1.2 V vs. RHE without the requirement of adscititious oxidant. Based on isotope labeling experiments and density functional theory (DFT) calculations, we deduce a possible reaction mechanism of selective photoelectrochemical oxidation of glycerol to DHA on $BiVO_4$.

## Results

**Synthesis and structural characterization of $BiVO_4$ photoanode**. $BiVO_4$ photoanode was prepared by reacting BiOI nanoflake arrays that were electrochemically grown on fluorine-doped tin dioxide (FTO) glass with vanadyl acetylacetonate at elevated temperatures. Supplementary Fig. 1 shows the typical scanning electron microscopy (SEM) image of the as-prepared $BiVO_4$ photoanode, which displays porous, interconnected nanoflake arrays. High-resolution transmission electron microscopy (HRTEM) image shows that the main exposed facet of the $BiVO_4$ nanoflake is (112). The $BiVO_4$ nanoarrays are phase-pure with monoclinic scheelite structure (Supplementary Fig. 2). The bandgap of the $BiVO_4$ nanoarrays was measured to be ~2.5 eV (Supplementary Fig. 3).

**Photoelectrochemical performance of $BiVO_4$ photoanode**. Photoelectrochemical (PEC) performance of $BiVO_4$ photoanode was first evaluated in 0.5 M $Na_2SO_4$ electrolyte with pH adjusted from 2 to 12 under one sun illumination (AM 1.5 G, 100 mW cm$^{-2}$). Figure 1 shows the current density-potential profiles under dark and illumination conditions. Without the addition of glycerol in the reaction medium, the photocurrent density resulting from water oxidation via front illumination is nearly identical, reaching ~1 mA cm$^{-2}$ at 1.23 V vs. RHE independent of pH. When glycerol was introduced, the photocurrent density dramatically increased, accompanying with a clear onset shift towards lower potentials, indicating easier oxidation of glycerol than water. The photocurrent

density approaches 4 mA cm$^{-2}$ at 1.23 V vs. RHE at pH = 2. Increase in pH greatly decreases photocurrent and at the same time increases the onset potential, suggesting that proton could directly influence the catalytic oxidation reaction. Further increasing pH to more than 10 caused photocorrosion of $BiVO_4$ (Supplementary Fig. 4).

Figure 1d–f and Supplementary Fig. 5 show the chopped photocurrent profiles recorded at 1.0 and 0.6 V vs. RHE, respectively in an H-type cell. In the absence of glycerol, each on–off cycle of photocurrent produces a transient spike, which is induced by the diffusion and accumulation of photogenerated holes at the surface of $BiVO_4$ due to the slow kinetics of water oxidation reaction (Supplementary Fig. 6 and Supplementary Note 1)[29]. The stable photocurrent density reaches 0.1 and 1.0 mA cm$^{-2}$ at 0.6 and 1.0 V vs. RHE, respectively at pH = 2. Adding 0.1 M of glycerol not only dramatically increases the photocurrent density to 1.8 and 3.4 mA cm$^{-2}$ at 0.6 and 1.0 V vs. RHE, but also at the same time greatly reduces the photocurrent spike. This comparison suggests faster reaction kinetics of glycerol oxidation as compared to water oxidation. Additionally, increase in reaction pH greatly reduces the photocurrent density, which can be attributed to the better glycerol adsorption on $BiVO_4$ at lower pH (Supplementary Fig. 7 and Supplementary Note 2). Enhanced adsorption of glycerol on $BiVO_4$ at lower pH shall result in easier transfer of photogenerated holes for further oxidation reactions, which therefore diminishes the chopped photocurrent spikes.

To gain information on the charge and mass transfer processes, electrochemical impedance spectroscopy (EIS) studies were performed under different applied biases with light illumination. An equivalent circuit based on charge carrier trapping mechanism was employed to analyze the EIS data (Fig. 2c)[30,31]. The equivalent circuit includes a space-charge capacitance, $C_{sc}$ and a surface state capacitance, $CPE_{\varphi}$. $R_{\Omega}$ is the series resistance, $R_t$ is the resistance that represents the trapping of holes in the surface states, and $R_s$ is the charge and mass transfer resistance from surface states to solution. For the EIS spectra collected without glycerol, as shown in the Bode (Fig. 2a) and Nyquist plots (Supplementary Fig. 8a), the frequency peak decreases and shifts to higher frequency with increasing bias, leading to decreased faradaic resistance and improved surface reaction rate. After adding glycerol, the frequency peak shifts to higher frequency, but the impedance semicircle becomes larger with increasing bias (Fig. 2b and Supplementary Fig. 8b). The increased $R_s$ in PEC glycerol oxidation with increasing bias could be attributed to the poor desorption of the oxidation products[32]. We further calculated the total resistance, $R_{total}$ via $R_{total} = R_{\Omega} + R_t + R_s$ (Supplementary Fig. 8c and Supplementary Note 3), and compared with the resistance estimated from the current density-potential profile ($R_{total} = dV\,dJ^{-1}$) as shown in Fig. 2d. At potentials prior to $H_2O$ oxidation onset, $R_{total}$ is found to be orders of magnitude higher for PEC water oxidation as compared to that for PEC glycerol oxidation.

To probe the charge carrier dynamics, we performed transient open-circuit voltage decay (OCVD) measurements. Figure 3a compares the OCVD profiles of the $BiVO_4$ photoanode measured in 0.5 M $Na_2SO_4$ aqueous solution at various pH, from which, we were able to estimate the lifetimes of photogenerated charge carriers (Fig. 3b). It is found that decrease in reaction pH slightly increases recombination rate of photogenerated electron–hole pairs. The photoresponse spectra of the porous $BiVO_4$ nanoarray photoanode were further assessed as shown in Fig. 3c, d, which resemble well with the light absorption spectrum of $BiVO_4$ (Supplementary Fig. 3). In the absence of glycerol at pH = 2, the incident photon-to-current conversion efficiency (IPCE) approaches ~20% at 400 nm and 1.23 V vs. RHE. The addition

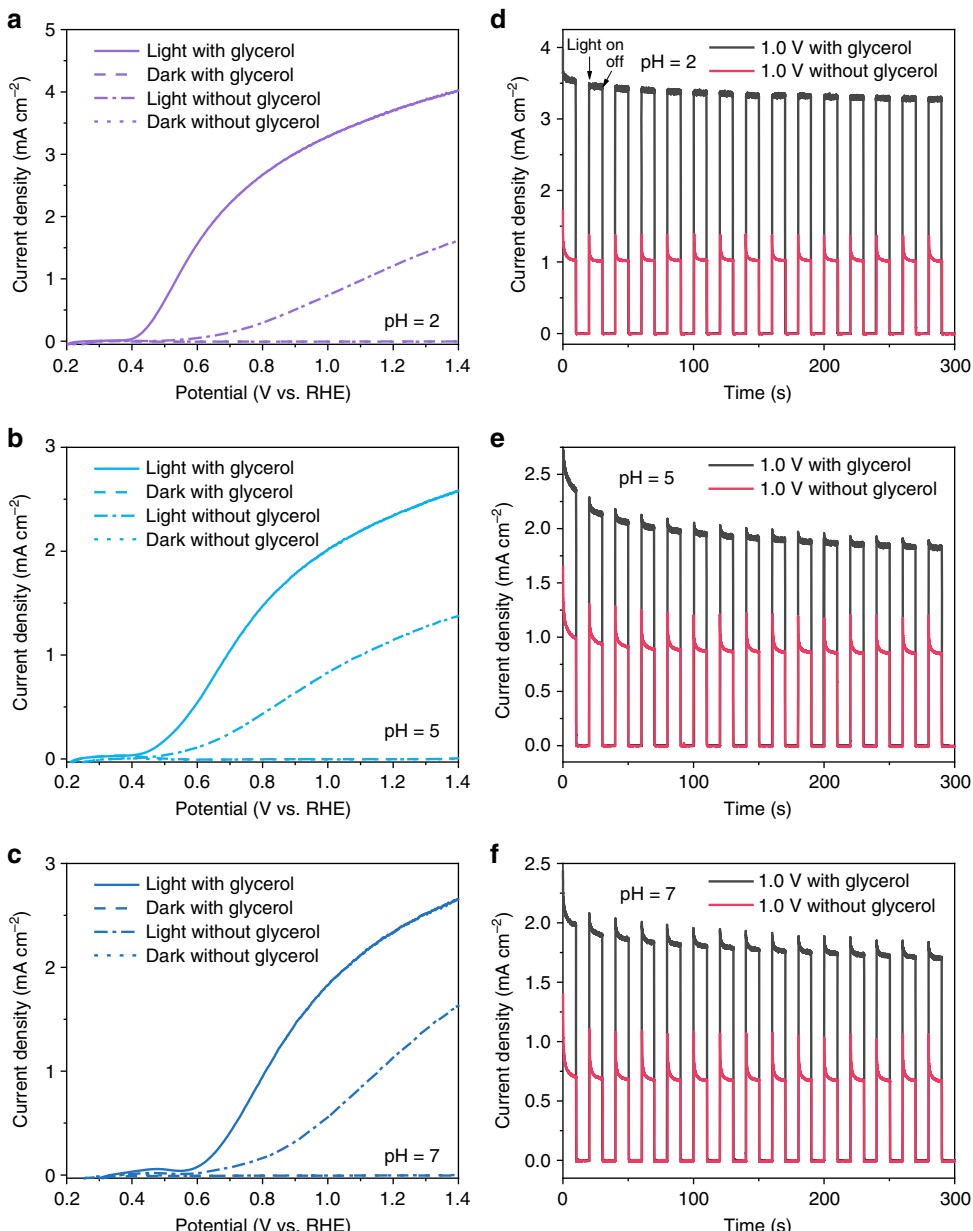

**Fig. 1** Photoelectrochemical performances. **a**–**c** Current density-potential profiles of BiVO$_4$ photoanode measured in 0.5 M Na$_2$SO$_4$ at various pH in an H-type cell under dark and AM 1.5 G, 100 mW cm$^{-2}$ illumination. **d**–**f** Chopped photocurrent density-time profiles of BiVO$_4$ photoanode at 1.0 V vs. RHE in 0.5 M Na$_2$SO$_4$ at various pH with and without glycerol. Source data are provided as a Source Data file

of glycerol significantly increases the IPCE across the entire light absorption region of BiVO$_4$, reaching 80% at 400 nm and 1.23 V vs. RHE, which is consistent with the change in photocurrent density. Furthermore, increase in reaction pH decreases IPCE (Fig. 3d) as a result of impaired binding of glycerol on BiVO$_4$ as discussed previously.

**Selective photoelectrochemical oxidation of glycerol**. The photoelectrochemical stability of the porous BiVO$_4$ nanoarray photoanode was assessed by measuring photocurrent as a function of reaction time at various applied potentials. As shown in Supplementary Fig. 9, the BiVO$_4$ photoanode exhibits excellent long-time stability against photocorrosion at pH = 2 with stable photocurrent densities. Following, the photooxidation of glycerol was carried out at 0.8–1.2 V vs. RHE to analyze the oxidation products and faradaic efficiency (Supplementary

Fig. 10). The products consist of formic acid, glyceric acid, DHA, and trace amount of glycolic acid. Figure 4a and Supplementary Fig. 11 show the production rate and selectivity of oxidation products at various pH. With increase in applied potential, the production rate of the oxidation products also increases. At pH = 2, DHA becomes the preferred oxidization product with selectivity and overall faradaic efficiency approaching ~50% in the liquid products and ~30%, respectively, independent of applied potential. At a potential of 1.2 V vs. RHE, the production rate of DHA reaches as high as about 200 mmol per m$^2$ of illumination area in 1 h, and the total carbon balance is about 91.3%, in which the glycerol-to-DHA conversion selectivity is 63.6% (Supplementary Table 1). CO$_2$ and CO could also be detected from further oxidation of liquid products together with H$_2$O$_2$ and O$_2$ from water oxidation. Total charge-to-chemical balance is about 90.21%. Besides DHA, glyceric acid was also produced as a by-product, resulting in the overall selectivity of

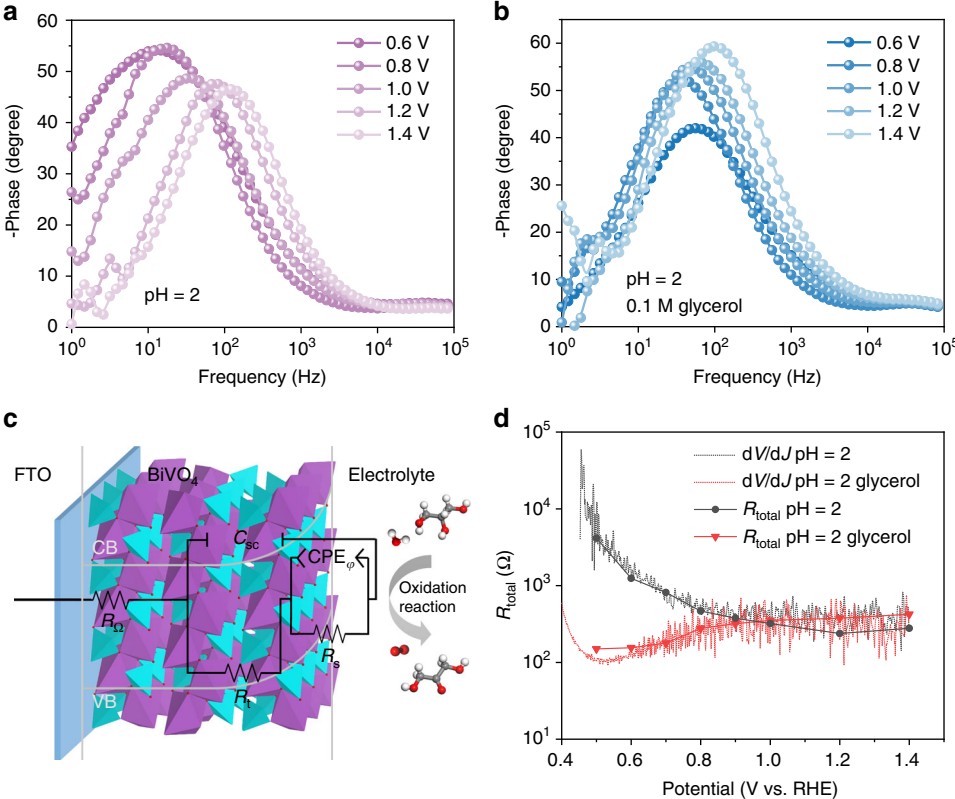

**Fig. 2** Charge and mass transfer processes. **a**, **b** The Bode plots measured at different potentials under AM 1.5 G, 100 mW cm⁻² illumination without and with the presence of glycerol. **c** Equivalent circuit used to conceptualize the BiVO₄ photoanode in PEC reactions. **d** $R_{total}$ calculated from $dV \, dJ^{-1}$ of current density-potential profiles and from EIS data with and without glycerol under illumination

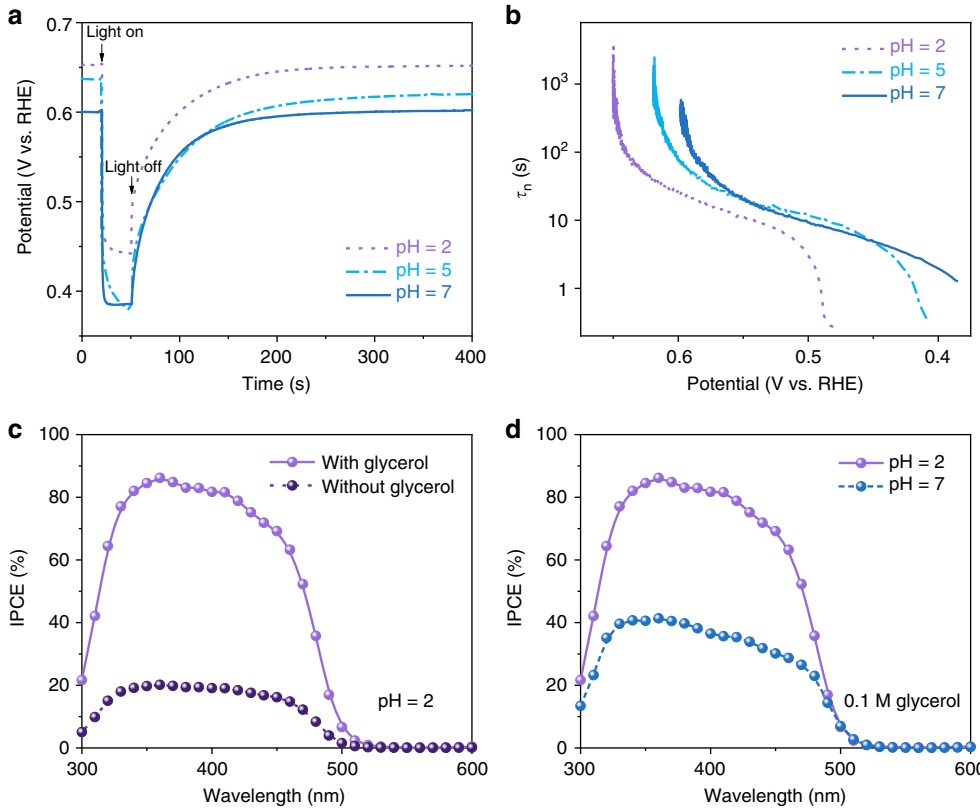

**Fig. 3** Charge carrier dynamics. **a**, **b** Transient OCVD profiles and lifetimes of photogenerated charge carriers of porous BiVO₄ nanoarray photoanode in 0.5 M Na₂SO₄ at various pH. **c**, **d** Incident photon-to-current conversion efficiency spectra of porous BiVO₄ nanoarray photoanode

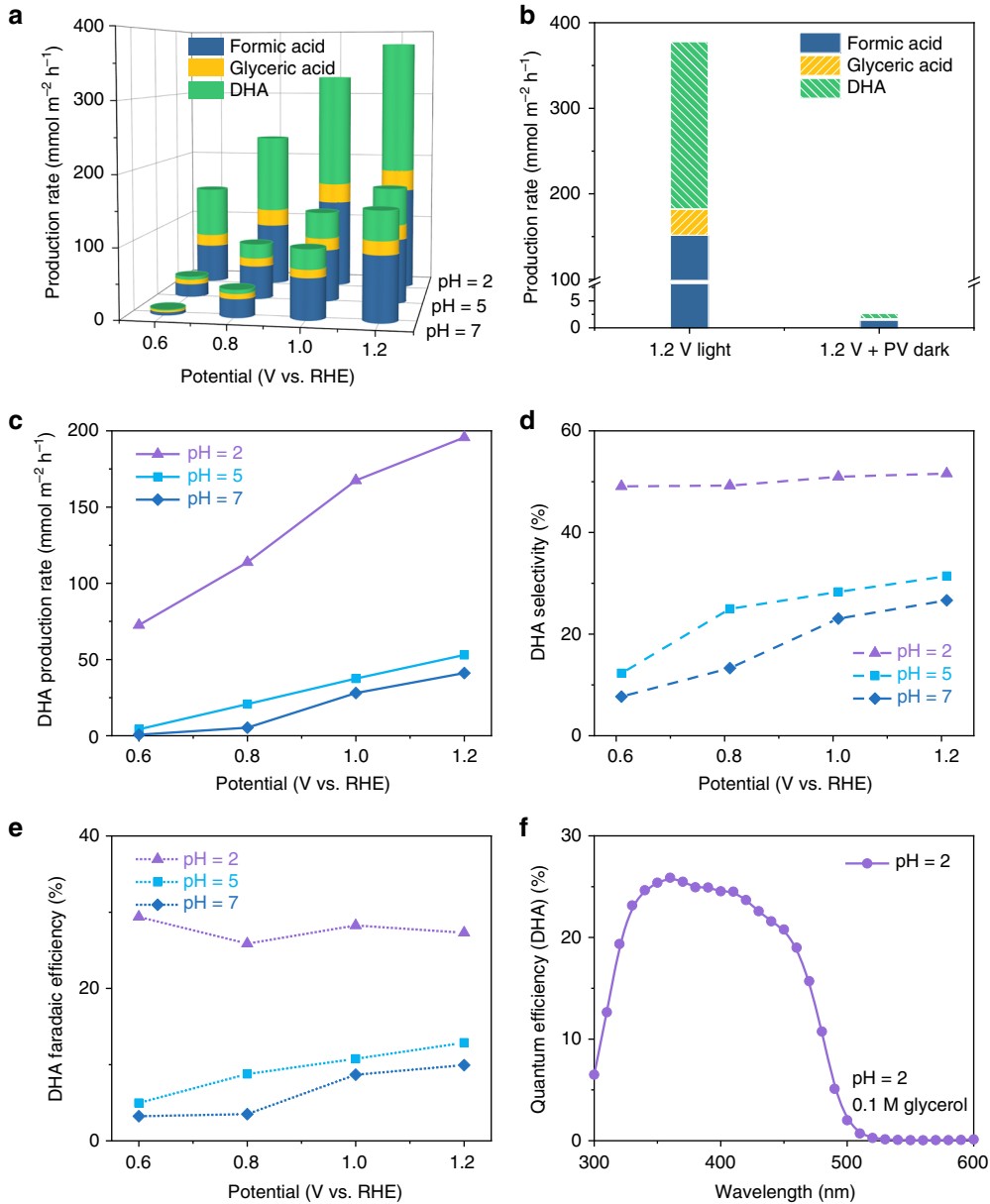

**Fig. 4** Selective photoelectrochemical oxidation of glycerol. **a** Photoelectrocatalytic production rate of oxidation products on BiVO$_4$ photoanode at reaction pH of 2, 5, and 7, respectively in 1 h. **b** Production rate of oxidation products at pH = 2 and 1.2 V vs. RHE under illumination or 1.375 V vs. RHE in dark, respectively **c–e** Production rate, selectivity, and faradaic efficiency of DHA produced in an H-type cell at pH = 2, 5, and 7. **f** Incident photon-to-DHA conversion efficiency spectrum acquired at pH = 2 and 1.23 V vs. RHE

high value-added glycerol derivatives (DHA + glyceric acid) approaching ~60%. Increase in pH decreases not only the photocurrent, but also the selectivity towards high value-added oxidation products (Fig. 4c–e). At pH = 12, glycerol was almost completely oxidized to formic acid with a faradaic efficiency close to 99% (Supplementary Fig. 12). Both photocurrent and production rate of DHA increase notably with increase of glycerol concentration (Supplementary Fig. 13). Without light illumination, glycerol could only be oxidized to DHA by BiVO$_4$ at very low efficiency after applying additional bias (Fig. 4b and Supplementary Fig. 14). Porous BiVO$_4$ nanoarray photoanode could produce DHA with higher rate and selectivity than BiVO$_4$ film photoanode as porous nanoarrays possess larger surface area and more abundant active sites (Supplementary Fig. 15 and Supplementary Note 4). In addition, porous BiVO$_4$ nanoarrays also showed advantages in selective glycerol oxidation as

compared to other semiconductor nanomaterials (Supplementary Fig. 16 and Supplementary Note 5).

**Reaction mechanism**. To probe the reaction mechanism, we conducted isotope-labeling experiments and electron spin resonance (ESR) measurements. Supplementary Fig. 17 displays the liquid chromatography mass spectrometry (LC-MS) results of photoelectrochemical glycerol oxidation in oxygen isotope labeled electrolyte with H$_2$O containing 10% H$_2{}^{18}$O. The results show that $^{18}$O can be detected in the target product DHA in the form of C$_3$H$_6$O$_2{}^{18}$ONa (Supplementary Table 2 and Supplementary Note 6), indicating that the isotope oxygen in DHA comes from water in the electrolyte. It can thus be speculated that oxidation of β-hydroxy to carbonyl group during photoelectrochemical glycerol oxidation to DHA would be accomplished by an addition

and substitution reaction. The adsorbed glycerol on BiVO$_4$ photoanode would be oxidized into radicals by photogenerated holes (Supplementary Tables 3, 4 and Supplementary Notes 7, 8)[33–35]. Room temperature ESR spectra indicate that after adding glycerol in this system, instead of hydroxyl radicals, glycerol radicals could be detected (Supplementary Fig. 18 and Supplementary Note 9). Driven by the energetic holes, the glycerol radicals could react with water from electrolyte to form unstable gem-diol intermediates, followed by dehydration to produce the carbonyl group, during which the isotope oxygen might be left in the products[35,36]. As a conclusion, we infer that the oxidation of glycerol to DHA could occur via the following steps as shown in Supplementary Fig. 19 and Supplementary Note 10, including adsorption of glycerol on BiVO$_4$ photoanode, oxidation of adsorbed glycerol to radicals by photogenerated holes, reaction with water, and dehydration. Glyceric acid and formic acid could also be formed via this similar pathway, resulted from the further oxidation of glyceraldehyde.

As shown in Supplementary Fig. 7, glycerol could be better adsorbed on BiVO$_4$ at lower pH, which shall result in an easier charge transfer and enhanced catalysis to photoelectrochemically convert glycerol into glycerol derivatives when decreasing reaction pH. Besides, acidic environment can also suppress the reaction towards acid products, which might further reduce the consumption of DHA as well as suppress the production of formic acid[37]. The adsorption of glycerol on BiVO$_4$ and the reaction pathways of the glycerol oxidation were verified by DFT calculations on the (112) facets of BiVO$_4$—the main exposed crystal facets of the BiVO$_4$ photoanode studied in this work (Supplementary Fig. 1c). As shown in the DFT results, the terminal and middle hydroxyl groups of glycerol adsorb spontaneously on the bismuth site of BiVO$_4$ with strong electrostatic attraction between Bi$^{3+}$ and oxygen of hydroxyls (Fig. 5, Supplementary Fig. 20, and Supplementary Note 11). Subsequently, the adsorbed glycerol is oxidized by the photo-generated holes in BiVO$_4$ to form radicals (Supplementary Figs. 21 and 22). The tertiary radical on the middle carbon is 0.18 eV more stable than the terminal radical, which explains the good selectivity towards DHA production during photoelectrochemical oxidation of glycerol on BiVO$_4$. The photo-assisted reaction is critical for the production of DHA as only formic acid could be detected in the final liquid product under dark condition. BiVO$_4$ provides a suitable bandgap of ~2.5 eV that endows photo-generated holes with sufficient energy to overcome the energy requirement for the formation of radicals, which is ~1.5 eV. Thereafter, the radicals react with water, followed by dehydration to eventually form DHA.

Finally, incident photon-to-DHA quantum efficiency (QE-DHA) was evaluated by multiplying IPCE with faradaic efficiency of DHA as shown in Fig. 4f. In the reaction medium with pH = 2, a 25% photon-to-DHA conversion efficiency can be achieved at a wavelength of 400 nm.

## Discussion

In summary, we have demonstrated an efficient and stable pho-toelectrochemical system based on porous BiVO$_4$ nanoarrays for selective oxidation of glycerol to high value-added DHA. In an H-type cell at pH = 2, the porous BiVO$_4$ nanoarray photoanode is able to produce 56 mmol g$_{catalyst}^{-1}$ h$^{-1}$ of DHA at a potential of 1.2 V vs. RHE under AM 1.5 illumination (100 mW cm$^{-2}$). Based on isotope labeling and DFT calculations, we deduced a possible reaction mechanism of selective photoelectrochemical oxidation of glycerol to DHA on BiVO$_4$. This study not only provides an evidence of the reaction process of selective photoelectrochemical oxidation of glycerol to DHA, but also presents a broad research

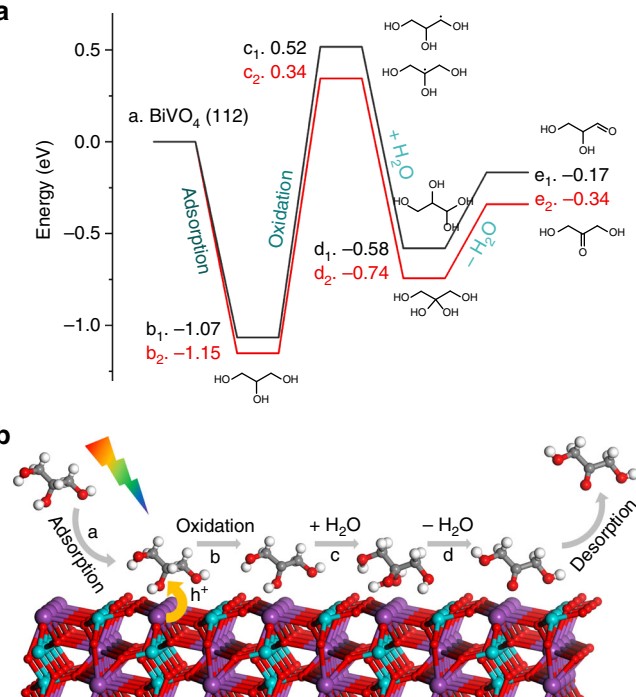

**Fig. 5** Reaction mechanism. **a** Energy profile of glycerol oxidation on BiVO$_4$. 1 (black) and 2 (red) stand for reactions that take place at the terminal and middle carbon, respectively. **b** Schematic illustration showing PEC glycerol oxidation to DHA

perspective for the utilization of solar light to produce high value-added chemicals from renewable feedstocks.

## Methods

**Synthesis of BiVO$_4$ photoanode.** All chemicals were purchased from Sigma-Aldrich and used without further purification. BiVO$_4$ photoanode was prepared by reacting BiOI nanoflake arrays with vanadyl acetylacetonate[38]. Firstly, BiOI nanoflake arrays were grown on conductive fluorine-doped SnO$_2$ (FTO, 14 Ω cm, 1 × 1 cm$^2$) by a three-electrode cathodic electrodeposition at −0.1 V vs. Ag/AgCl for 5 min in a mixed solution of 50 mL of 0.04 M Bi(NO$_3$)$_3$ and 0.4 M KI aqueous solution with pH adjusted to 1.7 by adding HNO$_3$ and 20 mL of 0.23 M benzo-quinone ethanol solution. Subsequently, 0.1 mL of vanadyl acetylacetonate dime-thyl sulfoxide solution (0.4 M) was added onto the BiOI nanoflake arrays, followed by heating to 723 K in ambient environment at a ramp rate of 2 K min$^{-1}$. After 2 h, the resultant product on FTO was cooled to room temperature and washed by NaOH solution.

**Sample characterizations.** SEM images were taken on a JEOL JSM-6700F field emission scanning electron microscope operated at 5 kV. HRTEM image was taken on a JEOL JEM-2100F field-emission transmission electron microscope operated at 200 kV. X-ray powder diffraction (XRD) patterns were recorded using a Philips X'Pert Pro Super X-ray diffractometer with Cu Kα radiation (λ = 1.54178 Å). UV–vis diffuse reflectance spectrum was recorded in the spectral region of 300–600 nm on a Shimadzu SolidSpec-3700 spectrophotometer. The bandgap of BiVO$_4$ was determined based on the Tauc plot. BiVO$_4$ photocatalysts used in this work were scraped from the BiVO$_4$ nanoarray-FTO glass.

Fourier transform infrared spectroscopy (FTIR) measurements were carried out in the attenuated total reflection (ATR) configuration using a Nicolet iN10 MX spectrometer equipped with a Ge ATR crystal and a DTGS KBr detector. All spectra were measured with a resolution of 1 cm$^{-1}$ and presented in absorbance $A = -\log(R/R_o)$, where $R$ and $R_o$ are the reflectance spectra of BiVO$_4$ samples treated in electrolyte of different pH with and without glycerol. After soaked in the electrolyte with and without glycerol at different pH for 10 min, the BiVO$_4$ samples were washed by the electrolyte without glycerol. After dried with N$_2$ flow, the BiVO$_4$ samples were pressed into intimate optical contact with the Ge crystal for ATR–FTIR measurements.

                                                                                                 

Room-temperature ESR spectra of $BiVO_4$ photocatalysts were collected on a JEOL JES-FA200 ESR spectrometer (300 K, 9.063 GHz, X-band). Microwave power employed was 1 mW and sweep width was set from 319.3 to 329.3 mT. Modulation frequency and modulation amplitude were 100 kHz and 0.35 mT, respectively. 5,5-Dimethyl-1-pyrroline N-oxide (DMPO) was used as the spin-trapping agent. 0.5 mg $BiVO_4$ photocatalysts were dispersed in 2 mL 0.5 M $Na_2SO_4$ electrolyte (pH = 2) with or without 0.1 M glycerol, then 60 μL DMPO aqueous solution (0.2 M) was added into the solution for ESR measurement.

**Photoelectrochemical characterization.** Photoelectrochemical measurements were conducted on an electrochemical workstation (CHI 760E) in a sealed H-type glass cell. Nafion 212 proton exchange membrane was used to separate the anode from the cathode chamber. The simulated solar illumination was obtained from a 300 W Xenon lamp with an AM 1.5G filter (100 mW cm$^{-2}$). The porous $BiVO_4$ nanoarrays grown on FTO were front illuminated and used as the working electrode, while Ag/AgCl electrode and Pt sheet were used as the reference and counter electrode, respectively. The electrolyte contains 0.5 M $Na_2SO_4$ aqueous solution with pH adjusted to 2, 5, 7, 10, and 12 using 1 M NaOH solution or 0.5 M $H_2SO_4$ solution. Photocurrent was recorded from 0.2 to 1.4 V vs. RHE at a scan rate of 10 mV s$^{-1}$ with and without the presence of 0.1 M glycerol.

The photoresponse of the $BiVO_4$ photoanode was studied by measuring photocurrent densities under chopped light irradiation (light on/off cycles: 20 s) at 0.6 and 1.0 V vs. RHE in 0.5 M $Na_2SO_4$ at various pH with and without glycerol. The photoelectrochemical impedance spectra were collected in a frequency range of 1~$10^5$ Hz with an amplitude of 5 mV under AM 1.5 G, 100 mW cm$^{-2}$ illumination, from 0.6 V to 1.4 V vs. RHE. Mott–Schottky plots were measured at 1000 Hz under dark and AM 1.5 G, 100 mW cm$^{-2}$ illumination with potential varied from −0.3 to 0.6 V vs. RHE.

OCVD measurements were carried out with light illumination on and off. After turning off the light, the average lifetime of photogenerated charge carriers ($\tau_n$) was estimated according to:

$$\tau_n = \frac{k_B T}{e} \left( \frac{dV_{oc}}{dt} \right)^{-1} \tag{1}$$

where $\tau_n$ is the potential dependent lifetime, $k_B$ is Boltzmann's constant, $T$ is the temperature in K, $e$ is the single electron charge, and $V_{oc}$ is the open-circuit voltage at time $t$.

IPCE was measured under monochromatic irradiation under one sun illumination (AM 1.5 G, 100 mW cm$^{-2}$) equipped with a monochromator at 1.23 V vs. RHE:

$$IPCE = \frac{\left[ \left( \frac{1240}{\lambda} \right) \times \left( J_{light} - J_{dark} \right) \right]}{P} \times 100\% \tag{2}$$

where $\lambda$ is the wavelength, $J_{light}$ is the photocurrent density under irradiation, $J_{dark}$ is the current density under dark condition, and $P$ is the incident light power density.

**Photoelectrocatalysis characterizations.** To quantitatively analyze the oxidation products, the photoelectrochemical oxidation was performed in a sealed H-type cell at a potential of 0.6, 0.8, 1.0, and 1.2 V vs. RHE for 1 h without the addition of adscititious oxidant. After reaction, 1 mL of solution was taken out from the cell and analyzed using high performance liquid chromatography (HPLC) to calculate the glycerol conversion and yields of the products using an Agilent 1260 chromatograph equipped with a Hi-Plex H (300×d-7 mm, 8 μm) column and MWD and RID detector. The external quantum efficiency of photon-to-DHA was calculated based on IPCE and faradaic efficiency of DHA.

Selectivity of DHA was calculated based on:

$$Selectivity(DHA) = \frac{n_{DHA}}{n_{all}} \times 100\% = \frac{n_{DHA}}{n_{GA} + n_{DHA} + n_{FA} + n_{GlA} + n_{LA}} \times 100\%$$
$$= \frac{c_{DHA}}{c_{GA} + c_{DHA} + c_{FA} + c_{GlA} + c_{LA}} \times 100\% \tag{3}$$

where $n_{DHA}$, $n_{GA}$, $n_{FA}$, $n_{GlA}$, and $n_{LA}$ are the yield of DHA, glyceric acid, formic acid, glycolic acid, and lactic acid, respectively. $c$ is the product concentration detected by HPLC.

Production rate of DHA was calculated according to:

$$Production\ rate(DHA) = \frac{c_{DHA} \times V}{t} \tag{4}$$

$$Production\ rate\ per\ unit\ area(DHA) = \frac{c_{DHA} \times V}{t \times A} \tag{5}$$

where $V$ is the volume of reaction solution, $t$ is the reaction time, and $A$ is the area of photoanode.

Faradaic efficiency was calculated by:

$$Faradaic\ efficiency(DHA) = \frac{Number\ of\ holes\ to\ oxidize\ glycerol\ to\ DHA}{Number\ of\ all\ collected\ photogenerated\ holes} \times 100\%$$
$$= \frac{e_{DHA} \times n_{DHA} \times N}{Q/n} \times 100\% \tag{6}$$
$$= \frac{2 \times c_{DHA} \times V \times N}{Q/n} \times 100\%$$

where $e_{DHA}$ is the number of holes required to oxidize one glycerol molecule to DHA, $N$ is Avogadro's constant, $Q$ is the quantity of electric charge, and $n$ is the elementary charge.

External quantum efficiency of DHA was calculated by:

$$QE(DHA) = \frac{Number\ of\ holes\ to\ oxidize\ glycerol\ to\ DHA}{Number\ of\ incident\ photons} \times 100\%$$
$$= IPCE \times Faradaic\ efficiency(DHA) \times 100\% \tag{7}$$

Carbon balance was calculated based on:

$$Carbon\ balance = \frac{n_{GA} + n_{DHA} + \frac{n_{FA}}{3} + \frac{n_{GlA}}{3/2} + \frac{n_{CO2}}{3} + \frac{n_{CO}}{3}}{n_{glycerol}} \times 100\% \tag{8}$$

The LC-MS measurements were carried out on a LTQ Orbitrap XL Hybrid Ion Trap-Orbitrap Mass Spectrometer with IonMax ESI (electrospray ionization) Source. 0.5 M of $Na_2SO_4$ solution containing 10% $H_2^{18}O$ was used as the electrolyte with pH adjusted to 2 by 0.5 M of $H_2SO_4$. After 2 h of PEC reaction, the products with electrolyte were collected. Before LC-MS measurements, the products were extracted by SPE (Solid Phase Extraction Cartridges) column to remove inorganic salts. Hypersil GOLD column (3 μm particle size) and ESI Na$^+$ source were used in the LC-MS measurements.

**Computational details.** All DFT calculations were performed with VASP 5.3.5[39,40]. Spin-polarized DFT with Perdew–Burke–Ernzerhof (PBE) flavor of generalized gradient approximation were employed[41]. Iterative solutions of the Kohn–Sham equations were expanded in plane-wave basis sets defined by a kinetic energy cutoff of 400 eV. The convergences of energy and force were set as $10^{-6}$ eV and 0.02 eV Å$^{-1}$, respectively. Single gamma point sampling was used for the Brillouin Zone integration. The $BiVO_4(112)$ surface was modeled by a p($2 \times 2$) supercell with three $VO_4$ layers as shown in Supplementary Fig. 20a. The calculated interlayer distance is 3.10 Å, which is in good agreement with HRTEM results. A vacuum region of 15 Å was set between the periodically repeated slabs to avoid artificial interactions. The charge density differences were evaluated using the formula $\Delta\rho = \rho_{slab+ads} - \rho_{slab} - \rho_{ads}$, where $\rho_{slab}$, $\rho_{ads}$, and $\rho_{slab+ads}$ are the electron densities of $BiVO_4(112)$ slab, adsorbate, and the adsorption configuration, respectively.

## Data availability

All data supporting the findings in the article as well as the Supplementary Information files are available from the corresponding authors on reasonable request. The source data underlying Fig. 1 and Supplementary Figs. 5, 10a-c, and 18 are provided as a Source Data file.

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

## Acknowledgements
This work was supported by the funds from the Singapore Ministry of Education Academic Research Fund (AcRF) Tier 1: RG115/18, RG115/17 and RG9/17, Tier 2: MOE2016-T2-2-004, the National Key Projects for Fundamental Research and Development of China (2017YFA0207301, 2016YFA0202804), the Strategic Priority Research Programme of the Chinese Academy of Sciences (XDB17020400), and the National Nature Science Foundation of China (21725102).

## Author contributions
D.L. and B.L. conceived the project and designed the experiments. B.L. supervised the project. D.L. carried out experiments and data analysis. J.-C.L., H.X., and J.L. performed numerical calculation. W.C., J.M., and H.B.Y. contributed to catalyst characterization. Y. X. and Y.H. contributed to the manuscript preparation. D.L. and B.L. together wrote the manuscript. All authors commented on the final manuscript.

## Additional information

**Competing interests:** The authors declare no competing interests.

