## [Peer Review File · Nature Communications]

Reviewers' comments:

Reviewer #1 (Remarks to the Author):

This manuscript describes a well known photoanode of BVO for oxidation of Glycerol. The control experiment of water oxidation on the prepared BVO indicates the catalytic property of the photoanode is similar to the old reported (eg. Science 2014, 991). In fact little worse than the old one). To use an alcohol as a hole scavenger is also well documented. Therefore the referee is struggling to find out the novelty of the materials design and the chemical process.

1. BVO can be photo-corroded (e.g. Nature Energy 2018,3, 53). The current work only carried out 1 h stability test, which is too short to draw a conclusion that the photoanode is stable in such a low pH ~2.
2. What is the overall faradaic efficiency on both anode and cathode? Is there any gaseous products eg. CO and CO₂ produced?
3. The reason behind the high selectivity was not convincing. What is the reaction pathway and what are the reaction intermediates proved experimentally?

Reviewer #2 (Remarks to the Author):

The authors report an interesting study on using photoelectrode (BiVO₄) to selectively oxidize glycerol to dihydroxyacetone. The main claimed novelty of the work is the high activity (as manifested by the high photocurrent) and the high selectivity (51%). The performance (mainly the selectivity) is indeed impressive, and for that, the work is publishable in Nat. Commun. However, the authors leave much to desire to improve the rigor of the current study. Major revisions are needed before the work can be finally published.

While I appreciate the value of the chosen reaction and the impressive performance, I gained little insight after reading the manuscript. For instance, what makes BiVO₄ unique? Is the porosity important? Is it critical to have light other than offering the photovoltage? On this point, the comparison as shown in Fig. 4b is meaningless. One should compare the products at 1.2 V under light and 1.2V+whatever photovoltage in dark. The DFT calculations as shown toward the end are nice, but they are little more than a handwaving argument. Little chemical insights are provided. Control experiments on other materials are needed to further strengthen the current study.

Reviewer #3 (Remarks to the Author):

This is an interesting and important work that reports a photoelectrochemical system based on nanoporous BiVO₄ for selective oxidation of glycerol to 1,3-dihydroxyacetone (DHA) with high selectivity, it may be accepted after considering the following comments.

- 1) The authors reported a selectivity of 51% to DHA, what is the faradaic efficiency (current efficiency) to DHA? Are there oxygen evolved from the system? The authors should report both selectivity and FE. Further, the authors reported four products from glycerol oxidation- glyceric acid, glycolic acid, formic acid, and DHA, were there other products examined? What are the carbon balance and charge-chemical balances in their work?
- 2) Is the porous BiVO₄ unique for photoelectrochemical oxidation of glycerol? How important is the morphology on its photoelectrochemical oxidation performance? Will BiVO₄ thin film with less pores still work?

3) Have the photoelectrochemical oxidation conditions been optimized? Is 0.1M highest glycerol concentration? I assume all main experiments were conducted with 0.1M glycerol.

4) For Fig 3c, d (Incident photon-to-current conversion efficiency spectra of porous BiVO₄ nanoarray photoanode), is it the total current? If it is, the IPCE of the control experiment (without glycerol) shows OER is the side reaction. It would help if an oxygen sensor system can be used to quantify evolved oxygen amount and then used to check charge-to-material balance.

5) For the photoelectrochemical oxidation mechanism study, I really like their isotopic H¹⁸O₂ and DFT computation work. However, I do not think that the first step must be "adsorption of glycerol on BiVO₄ photoanode". It may be just a complex photoelectrochemical process occurred at the interface between solid semiconductor BiVO₄ and liquid electrolyte. Any spectroscopy evidence shows glycerol is chemically adsorbed on BiVO₄ via a "chemical bond of O-Bi"? A recent paper tried to clarify difference between electrochemical reduction and electrocatalytic reduction (Mechanisms of Furfural Reduction on Metal Electrodes: Distinguishing Pathways for Selective Hydrogenation of Bioderived Oxygenates, Chadderdon XT, et al. JACS, 2017, 139, 14120).

6) Finally, from the application point of view, is the production rate of 200 mmol DHA/ m² of illumination area/ hour has significant benefit over other technologies for preparation of DHA? Did the authors consider separation of DHA from the system, especially low concentration of 0.1M glycerol is used. It would be better to provide techno-economic analysis of the reported photoelectrochemical synthesis of DHA from glycerol.

Point-by-point responses to the reviewers' comments

We sincerely thank all reviewers for their valuable comments and suggestions, which will certainly help to improve the quality of our work. We also appreciate the editor for giving us the opportunity to address the issues that we have failed to point out in the first version. Our point-by-point responses are presented below and all changes made in the revised manuscript have been highlighted in yellow.

Reviewer #1 (Remarks to the Author):

This manuscript describes a well know photoanode of BVO for oxidation of Glycerol. The control experiment of water oxidation on the prepared BVO indicates the catalytic property of the photoanode is similar to the old reported (eg. Science 2014, 991). In fact little worse than the old one. To use an alcohol as a hole scavenger is also well documented. Therefore the referee is struggling to find out the novelty of the materials design and the chemical process.

Response: We appreciate the concern raised by the reviewer; however, the reviewer may have overlooked the novelty demonstrated in this work.

BiVO_4 is an excellent n-type semiconductor, which is often used as photoanode for water oxidation. This manuscript reports photoelectrochemical oxidation of glycerol to high value-added product – DHA with appreciable selectivity and production rate using BiVO_4 nanoarrays as photoanode. DHA is widely used in cosmetic, pharmaceutical, fine chemical and food industries, which costs US \$150 per kilogram while the price for crude and refined glycerol is only US \$0.11 and US \$0.66 per kilogram, respectively. A glycerol molecule has three hydroxyls, and generally the selective oxidation of the middle hydroxyl is much more difficult in comparison with the oxidation of terminal hydroxyls due to the structural feature and unfavorable dynamics. The oxidation of glycerol to DHA also faces the challenge of uncontrollable further oxidation of DHA to other products. Therefore, it is an important process both scientifically and economically to realize the selective oxidation of glycerol to DHA. Glycerol here plays a crucial role, which is far more important than just being a hole scavenger. Glycerol can adsorb on BiVO_4 in acidic electrolyte, and be stimulated to radicals by photogenerated holes. As photoanode, nanoporous BiVO_4 arrays possess large surface area and abundant active sites than BiVO_4 film. Compared with other n-type semiconductors, like TiO_2 and $\alpha\text{-Fe}_2\text{O}_3$, nanoporous BiVO_4 photoanode shows much better performance at producing DHA in selectivity and production rate due to its advantages of adsorption of glycerol and the appropriate energy band gap for activation of radicals. Selective reaction and valuable chemicals have been achieved in this photoelectrochemical glycerol oxidation process by nanoporous BiVO_4 photoanode.

The chemical process of selective photoelectrochemical oxidation of glycerol is studied in this work for the first time. This work not only performs a photoelectrochemical selective oxidation of glycerol to DHA with high production rate, but also studies the possible reaction mechanism of this process. We studied the reaction mechanism through IR spectra, isotope-labeling experiments, ESR measurements and DFT calculations, and deduced a possible reaction pathway. Glycerol radicals could be detected *in-situ* during reaction. Photogenerated holes led to the activation of glycerol radicals. Water indeed participated in oxidation of glycerol to DHA as demonstrated by the isotope-labeling experiments. DFT results show that the possible reaction pathway contains selective adsorption of glycerol on BiVO_4 , glycerol radical stimulation by photogenerated holes, the participation of water, and dehydration into DHA.

1. BVO can be photo-corroded (e.g. Nature Energy 2018, 3, 53). The current work only carried out 1 h stability test, which is too short to draw a conclusion that the photoanode is stable in such a low pH ~2.

Response: We appreciate the reviewer for the valuable comment. BiVO₄ as photoanode would suffer from photo-corrosion in water oxidation. As shown in Fig. R1, in the first 1 h, the photocurrent of BiVO₄ photoanode decreased by about 55 % for water oxidation, and the color of BiVO₄ photoanode changed distinctly. After adding glycerol, the photo-corrosion of BiVO₄ could be greatly suppressed. After 5 h reaction, the photocurrent of glycerol oxidation only decreased by about 5 %, and the appearance of BiVO₄ photoanode did not change.

Figure R1. Photocurrent density-time profiles of BiVO₄ photoanode at 1.2 V vs. RHE in 0.5 M Na₂SO₄ at pH = 2 with and without 1 M glycerol. Insets show the photographs of the BiVO₄ photoanodes after 5 h reaction with and without 1 M glycerol, respectively.

2. What is the overall faradaic efficiency on both anode and cathode? Is there no gaseous products eg. CO and CO₂ produced?

Response: We appreciate the questions raised by the reviewer. We used gas chromatography to detect the gaseous products on both electrodes, and HPLC and Merck MQuant Peroxide Test strips were used to detect hydrogen peroxide. CO₂, CO, H₂O₂ and O₂ were detected on the photoanode resulting from further oxidation of liquid products and water oxidation, while H₂ could be detected on the cathode from water reduction.

The overall faradaic efficiency on anode at pH = 2 and 1.2 V vs. RHE:

64.35 % for glycerol derivative liquid products + 2.58 % for CO₂ + 0.63 % for CO + 19.79 % for H₂O₂ + 2.86 % for O₂ = 90.21 %.

The overall faradaic efficiency on cathode at pH = 2 and 1.2 V vs. RHE: 96.57 % for H₂.

3. The reason behind the high selectivity was not convincing. What is the reaction pathway and what are the reaction intermediates proved experimentally?

Response: We appreciate the reviewer's valuable comments. To experimentally probe reaction intermediates in photoelectrochemical reaction is indeed challenging because of liquid reaction environment and short lifetime of reaction intermediates. Nevertheless, we conducted the following *in-situ* experiments to help us study the reaction mechanism of selective photoelectrochemical glycerol oxidation to DHA. The IR spectra indicate that glycerol can better adsorb on BiVO₄ at lower pH. DFT results show that the middle hydroxyl of glycerol can adsorb on BiVO₄ more stably than terminal hydroxyl groups, and the tertiary radical is also more stable. As a

result, DHA produced by oxidation of tertiary radicals could possess higher selectivity. Moreover, the photogenerated holes by BiVO_4 with suitable energy could oxidize glycerol to DHA properly, but would not further oxidize DHA to other less valuable products.

We used room temperature Electron Spin Resonance (ESR) spectroscopy to explore the reaction intermediates during reaction. BiVO_4 scraped from BiVO_4 nanoarrays was used as the photocatalyst.

Figure R2. Room-temperature ESR spectra in 0.5 M Na_2SO_4 at pH = 2. (a) ESR spectra with 0.1 M glycerol but without catalyst in solution. (b) ESR spectra of BiVO_4 without glycerol. (c) ESR spectra of BiVO_4 with 0.1 M glycerol. (d) ESR spectra of three samples after 90 s illumination. The ones with “♦” belong to the spin adduct attributable to hydroxyl radicals. The ones with “◇” are attributed to the DMPO-glycerol radicals adduct.

The ESR spectra show that without BiVO_4 , glycerol could not be activated into radicals. Without glycerol, BiVO_4 could oxidize water into hydroxyl radicals. After adding glycerol in the system, instead of hydroxyl radicals, glycerol radicals were detected, indicating that glycerol could be stimulated into radicals by BiVO_4 photocatalysts. The ESR spectrum of DMPO-glycerol radical adduct observed here is similar to that of the DMPO-alcohol radical adducts.^{R1} Based on ESR spectra, isotope-labeling experiments and DFT calculations, the possible reaction pathway of selective photoelectrochemical oxidation of glycerol is deduced in Scheme S1.

Reviewer #2 (Remarks to the Author):

The authors report an interesting study on using photoelectrode (BiVO_4) to selectively oxidize glycerol to dihydroxyacetone. The main claimed novelty of the work is the high activity (as manifested by the high photocurrent) and the high selectivity (51%). The performance (mainly the selectivity) is indeed impressive, and for that, the work is publishable in Nat. Commun. However, the authors leave much to desire to improve the rigor of the current study. Major revisions are needed before the work can be finally published.

While I appreciate the value of the chosen reaction and the impressive performance, I gained little insight after reading the manuscript. For instance, what makes BiVO_4 unique? Is the porosity important? Is it critical to have light other than offering the photovoltage? On this point, the comparison as shown in Fig. 4b is meaningless. One should compare the products at 1.2 V under light and 1.2V+whatever photovoltage in dark. The DFT calculations as shown toward the end are nice, but they are little more than a handwaving argument. Little chemical insights are provided. Control experiments on other materials are needed to further strengthen the current study.

Response: We greatly appreciate the reviewer's positive and valuable comments.

BiVO_4 is an excellent photoanode material for water oxidation. In the photoelectrochemical glycerol oxidation system, BiVO_4 also exhibits its distinctive advantages. The hydroxyl groups of glycerol could be adsorbed strongly on Bi^{3+} of BiVO_4 . The energy band gap of BiVO_4 is appropriately high enough to overcome the reaction requirement for formation of radicals. At the same time, the energy of photogenerated holes of BiVO_4 is moderate so that they would not further oxidize DHA into other less valuable products.

BiVO_4 film photoanode was prepared to study the effect of porosity on selective photoelectrochemical oxidation of glycerol to DHA. Fig. R3 shows that the photocurrent density of BiVO_4 film photoanode is higher than that of porous BiVO_4 nanoarray photoanode for water oxidation, but after adding glycerol, the photocurrent density of film is lower than that of porous nanoarray photoanode. With larger surface area, more adsorption and active sites can be exposed in porous BiVO_4 nanoarrays. Compared with BiVO_4 film photoanode, this porous BiVO_4 nanoarray photoanode could produce DHA with higher selectivity and production rate as shown in Table R1.

Figure R3. (a) Photocurrent density-potential profiles of BiVO_4 film photoanode measured in 0.5 M Na_2SO_4 at pH = 2 with and without 0.1 M glycerol under AM 1.5G, 100 mW/cm^2 illumination. (b) Photocurrent density-time curve of BiVO_4 film measured in 0.5 M Na_2SO_4 at pH = 2 at 1.2 V vs. RHE with 0.1 M glycerol under AM 1.5G, 100 mW/cm^2 illumination. (c) Comparison of photoelectrocatalytic production rate of liquid oxidation products at pH = 2 and 1.2 V vs. RHE by porous BiVO_4 nanoarray photoanode and BiVO_4 film photoanode.

Table R1

	DHA selectivity / %	DHA efficiency / %	DHA production rate / $\text{mmol h}^{-1} \text{m}^{-2}$
Porous BiVO_4 nanoarray	51.6	27.3	195.7
BiVO_4 film	38.4	11.6	80.5

Light is critical for activation of glycerol into radicals since the process of radical activation needs ~ 1.5 eV energy. As shown in Fig. R4 and Table R2, after just applying photovoltage (0.175 V), the current density and production rate of DHA are very low without light illumination.

Figure R4. (a) Transient OCVD profile of porous BiVO_4 nanoarray photoanode in $0.5 \text{ M Na}_2\text{SO}_4$ at $\text{pH} = 2$ with 0.1 M glycerol. (b) Current density-time curves of BiVO_4 photoanode measured in $0.5 \text{ M Na}_2\text{SO}_4$ at $\text{pH} = 2$ and 1.2 V vs. RHE under illumination or 1.375 V vs. RHE under dark, respectively. (c) Production rate of liquid oxidation products at $\text{pH} = 2$ and 1.2 V vs. RHE under illumination or 1.375 V vs. RHE under dark, respectively.

Table R2

	DHA selectivity / %	DHA efficiency / %	DHA production rate / $\text{mmol h}^{-1} \text{m}^{-2}$
1.2 V light	51.6	27.3	195.7
1.2 V + PV dark	~ 34.6	~ 22.9	~ 1.0

TiO_2 nanowire and $\alpha\text{-Fe}_2\text{O}_3$ nanoporous photoanodes were prepared and their photoelectrocatalytic glycerol oxidation performances are shown in Fig. S15. Poor selectivity and production rate of DHA were achieved on these two photoanodes.

Regarding the DFT calculations, they are not a handwaving argument but compose solid evidence from a theoretical perspective: the results confirm the selectivity toward DHA over the terminal-oxidized products by the radical pathway, and that the photogenerated holes in BiVO_4 with a band gap of 2.5 eV possess energies sufficiently large to overcome the highest barrier (i.e., the one associated with the formation of radical) along the energy profile for production of DHA.

The radical pathway dictates the selectivity for DHA in two regards: one is the intrinsic stability of tertiary radical that is higher than that of the terminal radicals; the other arises from the particular surface structure of BiVO_4 that binds the tertiary radical better, as shown in Figure R5. The radical imposes a local planar structure, and this results in different chemisorption configurations for the two kinds of radicals: the tertiary radical molecule prefers binding to surface with the two terminal OH groups, while the terminal radical molecule is

forced to bind the surface with one terminal OH group and the middle OH group. The Bi-Bi distance on the BiVO_4 surface is more compatible with the binding configuration of tertiary radical, as clearly illustrated in Fig. R5, and this further promotes the radical pathway towards production of DHA.

Figure R5. The local binding configurations of (a) the tertiary radical molecule and (b) the terminal radical molecule on the BiVO_4 surface.

Reviewer #3 (Remarks to the Author):

This is an interesting and important work that reports a photoelectrochemical system based on nanoporous BiVO₄ for selective oxidation of glycerol to 1,3-dihydroxyacetone (DHA) with high selectivity, it may be accepted after considering the following comments.

Response: We greatly appreciate the reviewer's positive and valuable comments.

1) The authors reported a selectivity of 51% to DHA, what is the faradaic efficiency (current efficiency) to DHA? Are there oxygen evolved from the system? The authors should report both selectivity and FE. Further, the authors reported four products from glycerol oxidation- glyceric acid, glycolic acid, formic acid, and DHA, were there other products examined? What are the carbon balance and charge-chemical balances in their work?

Response: We appreciate the reviewer's valuable comments.

Faradaic efficiencies of DHA under different reaction conditions are presented in Fig. 4e, S10 and S11. At pH = 2 and 1.2 V vs. RHE, the nanoporous BiVO₄ could produce DHA with a Faradaic efficiency at 27.3 %.

CO₂ and CO can be detected besides liquid products. The Faradaic efficiencies of CO₂ and CO at pH = 2 and 1.2 V vs. RHE are 2.58 % and 0.63 %, respectively.

Hydrogen peroxide and oxygen can also be detected from photoelectrochemical water oxidation. The Faradaic efficiency of H₂O₂ and O₂ at pH = 2 and 1.2 V vs. RHE are 19.79 % and 2.86 %, respectively.

Carbon balance was calculated based on:

$$\text{Carbon balance} = \frac{n_{GA} + n_{DHA} + \frac{n_{FA}}{3} + \frac{n_{GLA}}{3/2} + \frac{n_{CO_2}}{3} + \frac{n_{CO}}{3}}{n_{glycerol}} \times 100\%$$

Table R3 shows the production rate and consumption rate of these chemicals.

Table R3

	Glycerol	Glyceric acid	DHA	Formic acid	Glycolic acid	CO ₂	CO	Carbon balance	Charge-to-chemical balance
Consumption rate / mmol h ⁻¹ m ⁻²	307.8								
Production rate / mmol h ⁻¹ m ⁻²		30.7	195.7	151.6	0.9	7.9	2.0		90.21 %
Conversion selectivity / %		10.0	63.6	16.4	0.2	0.9	0.2	91.3 %	

One of the liquid by-products, hydrogen peroxide, is an oxidizing agent. To study its influence, 0.1 M or 3 mM H₂O₂ was added into 0.5 M Na₂SO₄ at pH = 2 with 0.1 M glycerol under stirring. Table R4 shows the products after 1 h light illumination at pH = 2.

Table R4

Production rate / $\mu\text{M h}^{-1}$	Glyceric acid	DHA	Formic acid	Glycolic acid	DHA selectivity / %
0.1 M H_2O_2	18.3	3.7	130.1	11.9	2.3
3 mM H_2O_2	3.3	0	13.4	0	0
BiVO_4 photoanode	122.7	782.8	606.5	4.9	51.6

H_2O_2 has the ability to oxidize glycerol but with very poor selectivity to DHA. The used concentration of adsorbed H_2O_2 at 0.1 M and 3 mM are about 170 times and 5 times larger than the one produced from water oxidation by BiVO_4 photoanode, respectively. The results show that the production rate of DHA by oxidation of H_2O_2 is negligible, and the production rates of formic acid and glyceric acid by oxidation of H_2O_2 are also quite low. Consequently, the by-product H_2O_2 from water oxidation by BiVO_4 photoanode would contribute very little to the oxidation of glycerol, especially to the production of DHA.

2) Is the porous BiVO_4 unique for photoelectrochemical oxidation of glycerol? How important is the morphology on its photoelectrochemical oxidation performance? Will BiVO_4 thin film with less pores still work?

Response: We appreciate the questions raised by the reviewer.

As shown in Fig. R3 and Table R1, porous BiVO_4 nanoarray photoanode could produce DHA with higher selectivity, efficiency and production rate than BiVO_4 film. After adding glycerol, the photocurrent density of porous nanoarray photoanode increased distinctly as compared to that of the film photoanode. Porous structure could expose larger surface area and more active sites for glycerol adsorption and activation. BiVO_4 film photoanode could oxidize glycerol into DHA, but with lower selectivity, efficiency and production rate than the porous one.

3) Have the photoelectrochemical oxidation conditions been optimized? Is 0.1M highest glycerol concentration? I assume all main experiments were conducted with 0.1M glycerol.

Response: We appreciate the reviewer's valuable suggestions.

The photoelectrochemical oxidation conditions have been optimized by adjusting pH, applied bias, light intensity and glycerol concentration. Moderate reaction conditions were chosen to study the reaction mechanism. 1 M, 0.1 M and 0.01 M glycerol were added into the photoelectrochemical system. At pH = 2 and 1.2 V vs. RHE with different concentration of glycerol, the selectivity, efficiency and production rate of DHA are presented in Fig. R6 and Table R5. The one with 0.1 M glycerol could produce DHA with high selectivity and conversion efficiency. With 1 M glycerol, BiVO_4 photoanode could produce DHA at a production rate only 1.3 times than the one with 0.1 M glycerol.

Figure R6. (a) Photocurrent density-potential profiles of porous BiVO₄ nanoarray photoanode measured in 0.5 M Na₂SO₄ at pH = 2 with 1 M, 0.1 M and 0.01 M glycerol under AM 1.5G, 100 mW/cm² illumination. (b) Photocurrent density-time curves of porous BiVO₄ nanoarray photoanode measured in 0.5 M Na₂SO₄ at pH = 2 at 1.2 V vs. RHE with 1 M, 0.1 M and 0.01 M glycerol under AM 1.5G, 100 mW/cm² illumination. (c) Photoelectrocatalytic production rate of liquid oxidation products at pH = 2 and 1.2 V vs. RHE by porous BiVO₄ nanoarray photoanode.

Table R5

	DHA selectivity / %	DHA efficiency / %	DHA production rate / mmol h ⁻¹ m ⁻²
0.01 M Glycerol	48.1	26.1	120.2
0.1 M Glycerol	51.6	27.3	195.7
1 M Glycerol	61.6	27.4	259.1

4) For Fig 3c, d (Incident photon-to-current conversion efficiency spectra of porous BiVO₄ nanoarray photoanode), is it the total current? If it is, the IPCE of the control experiment (without glycerol) shows OER is the side reaction. It would help if an oxygen sensor system can be used to quantify evolved oxygen amount and then used to check charge-to-material balance.

Response: We appreciate the questions raised by the reviewer.

Total current was measured for IPCE spectra in Fig. 3c and d. Besides glycerol oxidation, water oxidation also took place as a side reaction. Hydrogen peroxide can be detected by HPLC and Peroxide Test strips, together with little oxygen detected by GC. The charge balance is presented in Table R3. Please see our detailed response to the comment #1.

5) For the photoelectrochemical oxidation mechanism study, I really like their isotopic H₂¹⁸O and DFT computation work. However, I do not think that the first step must be “adsorption of glycerol on BiVO₄ photoanode”. It may be just a complex photoelectrochemical process occurred at the interface between solid semiconductor BiVO₄ and liquid electrolyte. Any spectroscopy evidence shows glycerol is chemically adsorbed on BiVO₄ via a “chemical bond of O-Bi”? A recent paper tried to clarify difference between electrochemical reduction and electrocatalytic reduction (Mechanisms of Furfural Reduction on Metal Electrodes: Distinguishing Pathways for Selective Hydrogenation of Bioderived Oxygenates, Chadderdon XT, et al. JACS, 2017, 139, 14120).

Response: We appreciate the reviewer’s valuable suggestions.

Adsorption of glycerol on BiVO₄ photoanode seems necessary for this photoelectrochemical reaction. We compared this glycerol oxidation reaction after adding hole scavenger, ammonium formate, and hydroxyl radical scavenger, *t*-butanol. The catalytic performance (Table R6) shows that after adding hole scavenger, DHA could not be detected, and meanwhile, after adding hydroxyl radical scavenger, the production of DHA was not affected too much. It indicates that holes should directly participate in the glycerol oxidation reaction instead of hydroxyl radicals.

Table R6

Production rate / mmol h ⁻¹ g ⁻¹	DHA	Formic acid	Glyceric acid	DHA selectivity / %
BiVO ₄ photocatalysts	0.138	0.144	0.072	39.0
+ 0.1 M ammonia formate	0	--	0.139	0
+ 0.1 M t -butanol	0.143	0.227	0.066	32.7

Room temperature ESR spectra as shown in Fig. R2 also reveal that glycerol would be activated into carbon radicals by photogenerated holes from BiVO₄. Unlike hydroxyl radicals, the migration distance of holes is very short, and it makes adsorption of glycerol necessary for this selective photoelectrochemical reaction. From the experimental results and DFT calculations, reaction pathway of photoelectrochemical glycerol oxidation contains adsorption of glycerol on BiVO₄, glycerol radicals' stimulation by photogenerated holes, middle gem-diol's formation from glycerol radicals and water, and dehydration of gem-diol into DHA. From the perspective of the proposed literature (JACS, 2017, 139, 14120), this photoelectrochemical glycerol oxidation process is likely the combination of "electrocatalytic" and "electroreduction" process.

It is hard to recognize the signal of chemical bond of O-Bi from the ATR-FTIR absorbance spectra since the nearby signals of Na₂SO₄, H₂SO₄ and water interfere the signals of glycerol. But DFT calculation results show that there exists strong electrostatic attraction between Bi³⁺ and O²⁻, and the distance and strength of Bi-O between layers of BiVO₄ and O(glycerol)-Bi(BiVO₄) are similar. The DFT calculations predict energies of the first step (i.e., adsorption of glycerol on BiVO₄) downhill by larger than 1 eV, and this enables strong spontaneous adsorption and thus naturally incorporates the adsorption into the photoelectrochemical oxidation process.

6) Finally, from the application point of view, is the production rate of 200 mmol DHA/ m² of illumination area/ hour has significant benefit over other technologies for preparation of DHA? Did the authors consider separation of DHA from the system, especially low concentration of 0.1M glycerol is used. It would be better to provide techno-economic analysis of the reported photoelectrochemical synthesis of DHA from glycerol.

Response: We appreciate the reviewer's valuable comments.

Production rate of DHA at about 200 mmol h⁻¹ m⁻² (56 mmol g_{catalyst}⁻¹ h⁻¹) achieved in this work is high compared with most catalytic methods for DHA production. Thermal oxidation and electrochemical oxidation on noble metal catalysts were used before for glycerol oxidation into DHA, which are not only cost-intensive, but also produce DHA with low production rates. Microbial fermentation process is another main DHA production method, which is complex, time-consuming, and subject to substrate inhibition. BiVO₄ used in this work can be prepared in large-scale at low cost. Driven by clean energy, this photoelectrochemical glycerol oxidation process is sustainable and economical. At pH = 2 and 1.2 V vs. RHE under AM 1.5, 100 mW/cm² illumination, the charge-to-DHA efficiency is as high as 27.3 %, incident photon-to-DHA quantum efficiency is about 25 %

at 400 nm, and glycerol-to-DHA conversion selectivity is as high as 63.6 %.

The physical properties of these liquid products are quite different, e.g., solubility, melting point and boiling point. Chromatography method and multi-effect evaporative crystallization method can be used for separation of DHA from this system.^{R2, R3}

References

- R1. Wu, W. *et al.* A new insight into the photocatalytic reduction of 4-nitroaniline to p-phenylenediamine in the presence of alcohols. *Appl. Catal. B Environ.* **130**, 163-167 (2013).
- R2. Lari, G. M. *et al.* Environmental and economic assessment of glycerol oxidation to dihydroxyacetone over technical iron zeolite catalysts. *React. Chem. Eng.* **1**, 106-118 (2016).
- R3. Ning, X. *et al.* Promoting role of bismuth and antimony on Pt catalysts for the selective oxidation of glycerol to dihydroxyacetone. *J. Catal.* **335**, 95-104 (2016).

Reviewers' comments:

Reviewer #1 (Remarks to the Author):

The authors did improve the technical presentation of the manuscript after revision. However it still lacks of solid evidence to prove the novelty. Authors underlined the significance of the product DHA which the referee agrees however it does not support the novelty of the chemical process as the current concentration used for DHA production is too low to talk about economic benefit. The referee would like to understand the scientific novelty, further the reason of the high selectivity and stability of the photoanode.

1) authors used isotopic water (H_2O containing 10% $H_2^{18}O$) as an electron donor to prove the proposed reaction pathway as suggested in Scheme 1. The referee would like to know i) why the peak intensity of $C_3H_6O_2^{18}ONa$ is so close to that of $C_3H_6^{16}O_3Na$ although there is 90% of $H_2^{16}O$ in the electrolyte and ii) is there peak attributed to $C_3H_6^{16}O^{18}O_2Na$ due to the fast exchange between ^{16}O and ^{18}O in the electrolyte?

2) how to rule out the possible pathway in which water is first oxidized to OH radicals that further oxidize glycerol to intermediates? Similar concern from other reviewer indicates the current pathway is one option.

3) Is the stability of the photoanode in glycerol oxidation affected by the concentration of the reactant as one can see the concentration of glycerol (in the response letter which is 1M) increases by a factor of 10 compared with the condition used in the Fig S9 (0.1M)? As authors claim the photoanode is stable which is a strong claim for Bi-based photocatalysts, the referee would suggest to add the measured stability over a few hours to the main context, eg. replacing the current fig 1d which only shows a 5min test. If one looks closely, there is clear difference of photocurrents represented in Fig 1a and FigS9a although both used identical condition.

4) If the measured faradic efficiency of BVO is not unity (at present 90%), does it indicate the photocorrosion of the photoanode takes place, which is again the big concern of the referee?

5) the last point is about selectivity. The referee does not follow the reason why the middle instead of terminal hydroxide is oxidized preferentially.

6) this claim of a production rate of 200 mmol of 1,3-dihydroxyacetone per m^2 of illumination area in one hour would cause misleading as the activity will not linearly increase with the surface area.

Reviewer #2 (Remarks to the Author):

I have gone through the authors' revisions and feel they have adequately addressed my concerns. The paper should be published without further delays.

Reviewer #3 (Remarks to the Author):

I am satisfied with the response of the authors to reviewers' comments. I endorse its publication in Nature Communications.

Point-by-point responses to reviewers' comments

We sincerely thank all reviewers for their careful review and valuable comments, which certainly help to improve the quality of our manuscript. Our point-by-point responses are presented below, and all changes made in the revised manuscript are highlighted in yellow.

Reviewer #1 (Remarks to the Author):

The authors did improve the technical presentation of the manuscript after revision. However, it still lacks solid evidence to prove the novelty. Authors underlined the significance of the product DHA which the referee agrees however it does not support the novelty of the chemical process as the current concentration used for DHA production is too low to talk about economic benefit. The referee would like to understand the scientific novelty, further the reason of the high selectivity and stability of the photoanode.

Response: We appreciate the concerns raised by the reviewer.

The chemical process and possible reaction mechanism of selective photoelectrochemical oxidation of glycerol was studied in this work. This photogenerated-hole-driven and glycerol radical catalysis process by BiVO₄ photoanode could produce valuable chemical — DHA with high production rate and selectivity. We studied the reaction mechanism through IR spectra, isotope-labeling experiments, ESR measurements and DFT calculations. Glycerol radicals could be detected *in-situ* during reaction by ESR spectra. Then we proved that photogenerated holes led to the activation of glycerol radicals by controlled experiments. Isotope-labeling results indicated that water molecules indeed participated in oxidation of glycerol to DHA. DFT results show that the possible reaction pathway contains selective adsorption of glycerol, glycerol radical stimulation by photogenerated holes, the participation of water, and dehydration into DHA.

The photocurrent density to produce DHA is about 1.3 mA cm⁻² at 1.2 V vs. RHE under 100 mW cm⁻² illumination in 0.5 M Na₂SO₄ at pH = 2 with 1 M glycerol. For solar-to-hydrogen conversion research, the solar-to-hydrogen (STH) efficiency target is ca. 10 % by US Department of Energy, equivalent to a current density of 8.13 mA cm⁻² under 100 mW cm⁻² light illumination.^{R1, R2} In terms of product value, DHA is much more valuable than hydrogen.^{R3, R4} Accordingly, this current density to produce DHA at 1.3 mA cm⁻² is very appreciable. This work may also induce more scientific research on optimizing this selective photoelectrochemical oxidation system to further improve the photocurrent and efficiency.

1) Authors used isotopic water (H₂O containing 10% H₂¹⁸O) as an electron donor to prove the proposed reaction pathway as suggested in Scheme 1. The referee would like to know i) why the peak intensity of C₃H₆O₂¹⁸ONa is so close to that of C₃H₆¹⁶O₃Na although there is 90% of H₂¹⁶O in the electrolyte and ii) is there peak attributed to C₃H₆¹⁶O¹⁸O₂Na due to the fast exchange between ¹⁶O and ¹⁸O in the electrolyte?

Response: We appreciate the questions raised by the reviewer.

i) The products of glycerol oxidation with electrolyte were first purified using SPE (solid phase extraction) column for several times before LC-MS measurements. DHA is a solid chemical at normal temperatures. It might be extracted some together with electrolyte by SPE column. LC-MS was used to detect the mass spectra of liquid molecules. The working temperature of ion transfer tubes is 275 °C in this LC-MS method with ESI Na⁺ source. At this high temperature, it is very easy for glycerol and DHA to be carbonized since they both have carbon skeletons of three carbon atoms. Consequently, these LC-MS results can only be used to analyze the chemical process qualitatively, not quantitatively. ii) The peak attributed to C₃H₆¹⁶O¹⁸O₂Na was not detected.

2) how to rule out the possible pathway in which water is first oxidized to OH radicals that further oxidize glycerol to intermediates? Similar concern from other reviewer indicates the current pathway is one option.

Response: We appreciate the questions raised by the reviewer.

ESR results in Fig. S18 in the supplementary information show that after adding glycerol, glycerol radicals could be detected instead of hydroxyl radicals under light illumination. It indicates that glycerol could be stimulated into radicals by BiVO₄ photocatalysts. The experimental results in Table S3, and ESR and DFT results could all show that photogenerated holes directly participate in the glycerol oxidation reaction and are crucial for this glycerol-DHA selective conversion.

As a contrast, hydroxyl radicals were introduced in this glycerol photocatalysis system by Fenton reaction.^{R5, R6} 0.3 M H₂O₂ and 3 mM Fe₂SO₄ were added into 0.5 M Na₂SO₄ at pH = 2 with 0.1 M glycerol under stirring. As shown in Table R1, hydroxyl radicals would oxidize glycerol with or without BiVO₄ photocatalysts under dark or 100 mW/cm² illumination. The selectivity of DHA by oxidation of OH radicals is as low as 13~14 %. The results indicate that this homogeneous glycerol oxidation process driven by hydroxyl radicals could only produce DHA with low selectivity. On the other hand, it also shows that it is harder for middle hydroxyls of glycerol to be oxidized than terminal hydroxyls.

Consequently, OH radicals do not play a crucial role in the photoelectrochemical glycerol oxidation process on BiVO₄. Instead, it is photogenerated holes, not OH radicals that oxidize glycerol into radicals with high selectivity.

Table R1

Production rate / $\mu\text{M h}^{-1}$	Glyceric acid	DHA	Formic acid	Glycolic acid	DHA selectivity / %
dark	254	494	2763	11	14.0
light	570	1127	6840	21	13.2
dark with BiVO ₄ nanocatalysts	309	508	2908	10	13.6
light with BiVO ₄ nanocatalysts	698	1681	9623	29	13.9

3) Is the stability of the photoanode in glycerol oxidation affected by the concentration of the reactant as one can see the concentration of glycerol (in the response letter which is 1M) increases by a factor of 10 compared with the condition used in the Fig S9 (0.1M)? As authors claim the photoanode is stable which is a strong claim for Bi-based photocatalysts, the referee would suggest to add the measured stability over a few hours to the main context, eg. replacing the current fig 1d which only shows a 5min test. If one looks closely, there is clear difference of photocurrents represented in Fig 1a and FigS9a although both used identical condition.

Response: We appreciate the reviewer's valuable comments.

The concentration of the reactant will influence the photocurrent of BiVO₄ photoanode as shown in Fig. S12. The concentration of glycerol decreases as the reaction proceeds, and consequently the photocurrent would continuously decrease during photoelectrochemical oxidation of glycerol. However, this photocurrent decrease does not mean that the BiVO₄ photoanode is not stable, but is resulted from decrease of the glycerol concentration in the reaction system. As shown in Fig. R1, the photocurrent of glycerol oxidation only decreased a little bit after 5 h reaction. We have added the figure in the revised Supplementary Information (Fig. S9). Fig.

1d-f and Fig. S5 in the manuscript show the chopped photocurrent profiles, which indicate that photoelectrochemical glycerol oxidation has faster reaction kinetics as compared to water oxidation.

Figure R1. Photocurrent density-time profiles of BiVO₄ photoanode at 1.2 V vs. RHE in 0.5 M Na₂SO₄ at pH = 2 with and without 1 M glycerol. Insets show the photographs of the BiVO₄ photoanodes after 5 h reaction with and without 1 M glycerol, respectively.

Without glycerol, the photocurrent of BiVO₄ photoanode decreased a lot due to photo-corrosion or self-oxidation of BiVO₄ as shown in Fig. R1. After adding glycerol, the photocurrent stability was significantly improved since photogenerated holes would oxidize abundant glycerol molecules instead of BiVO₄ itself. The SEM image and XRD pattern of BiVO₄ photoanode after 1 h photoelectrochemical glycerol oxidation are shown in Fig. R2. It is clear to see that BiVO₄ photoanode still maintains stable crystal structure and nanoporous array morphology.

Figure R2. (a) Top-view SEM image of BiVO₄ nanoporous arrays after 1 h photoelectrochemical reaction. (b) XRD patterns of BiVO₄ nanoporous arrays before and after reaction.

The difference of the photocurrents in Fig. 1a and S9a in the manuscript is because of the little individual differences of BiVO₄ photoanodes in different experiments. After adding error bars, the photocurrent of BiVO₄ photoanode in 0.5 M Na₂SO₄ at pH = 2 with 0.1 M glycerol is shown in Fig. R3.

Figure R3. Photocurrent density-potential profile of BiVO₄ photoanode measured in 0.5 M Na₂SO₄ at pH = 2 with 0.1 M glycerol under AM 1.5G, 100 mW/cm² illumination with error bars.

4) If the measured faradic efficiency of BVO is not unity (at present 90%), does it indicate the photocorrosion of the photoanode takes place, which is again the big concern of the referee?

Response: We appreciate the reviewer's valuable comments.

As shown in Table R3, the total carbon balance is about 91.3 % at pH = 2 and 1.2 V vs. RHE. And the overall faradaic efficiency on cathode at pH = 2 and 1.2 V vs. RHE is also not unity (about 96.6 %). It indicates that there might be a very small amount of by-products we could not identify due to small unidentified peaks or the limit of detection sensitivity within the margin of error. Photo-corrosion of photoanode might happen during long-time high-current photoelectrochemical reactions. On the other hand, gas tightness of the H-type cell could influence the quantification of gas products on both electrodes. In general, most products can be detected, and the charge balance and carbon balance both approach unity.

5) the last point is about selectivity. The referee does not follow the reason why the middle instead of terminal hydroxide is oxidized preferentially.

Response: We appreciate the reviewer's valuable comments.

The selective adsorption of glycerol on BiVO₄ is the first step of this reaction. Due to the strong electrostatic attraction between Bi³⁺ and oxygen of hydroxyls, the terminal and middle hydroxyl groups of glycerol can adsorb spontaneously on the bismuth site of BiVO₄. The adsorption of middle hydroxyl groups of glycerol on BiVO₄ is 0.06 eV more stable than the terminal hydroxyl groups as shown in Fig. 5, which will lead to preferential adsorption of middle hydroxyl groups on BiVO₄. Then the photogenerated holes will transfer and oxidize the adsorbed glycerol into radicals. The tertiary radicals on the middle carbon is 0.18 eV more stable than the terminal radicals. Following by reaction with water and dehydration, the middle carbon radicals still possess preferable energy kinetics. Unlike oxidation by hydroxyl radicals, this photogenerated hole-driven oxidation process could produce DHA with high selectivity.

6) this claim of a production rate of 200 mmol of 1,3-dihydroxyacetone per m² of illumination area in one hour would cause misleading as the activity will not linearly increase with the surface area.

Response: We appreciate the reviewer's valuable comments.

As shown in Fig. R4 and Table R2, the normalized production rates of DHA by BiVO₄ photoanode with different illumination areas are all around 200 mmol m⁻² h⁻¹ (herein, the area refers to the illumination area). For this type of nanoporous BiVO₄ array photoanode, the production rate of DHA nearly linearly increases with the

illumination area of photoanode (the production rate of DHA normalized to illumination area keeps nearly constant).

Figure R4. DHA production rate of BiVO₄ photoanode with different illumination area in 0.5 M Na₂SO₄ at pH = 2 with 0.1 M glycerol under AM 1.5G, 100 mW/cm² illumination.

Table R2

Illumination area / cm ²	DHA production rate / µM h ⁻¹	DHA normalized production rate / mmol m ⁻² h ⁻¹	DHA selectivity / %
0.58	487	210	53.0
1	783	196	51.6
1.5	1246	201	53.2

Reviewer #2 (Remarks to the Author):

I have gone through the authors' revisions and feel they have adequately addressed my concerns. The paper should be published without further delays.

Response: We greatly appreciate the reviewer's positive and valuable comments. We warmly thank the reviewer for recommending our work for publication in Nature Communications.

Reviewer #3 (Remarks to the Author):

I am satisfied with the response of the authors to reviewers' comments. I endorse its publication in Nature Communications.

Response: We greatly appreciate the reviewer's positive and valuable comments. We warmly thank the reviewer for recommending our work for publication in Nature Communications.

References

- R1. Esposito, D. V., Levin, I., Moffat, T. P. & Talin, A. A. H₂ evolution at Si-based metal-insulator-semiconductor photoelectrodes enhanced by inversion channel charge collection and H spillover. *Nat. Mater.* **12**, 562-568 (2013).
- R2. DOE Technical Targets for Hydrogen Production from Photoelectrochemical Water Splitting. <https://www.energy.gov/eere/fuelcells/doe-technical-targets-hydrogen-production-photoelectrochemical-water-splitting>
- R3. Kumar, G. S. *et al.* Stabilized glycerol dehydrogenase for the conversion of glycerol to dihydroxyacetone. *Chem. Eng. J.* **276**, 283-288 (2015).
- R4. Lee, D. Y., Elgowainy, A. & Dai, Q. Life cycle greenhouse gas emissions of hydrogen fuel production from chlor-alkali processes in the United States. *Appl. Energ.* **217**, 467-479 (2018).
- R5. Gligorovski, S., Streckowski, R., Barbati, S. & Vione, D. Environmental implications of hydroxyl radicals ($\bullet\text{OH}$). *Chem. Rev.* **115**, 13051-13092 (2015).
- R6. Zepp, R. G., Faust, B. C. & Hoigne, J. Hydroxyl radical formation in aqueous reactions (pH 3-8) of Iron(II) with hydrogen-peroxide: the photo-Fenton reaction. *Environ. Sci. Technol.* **26**, 313-319 (1992).

Reviewers' comments:

Reviewer #1 (Remarks to the Author):

The referee has a few questions, which have not been properly addressed in the revised.

1) These IR spectra, isotope-labeling experiments and ESR measurements do shine light on the possible reaction pathway, while cannot disclose the key information related to the selectivity to DHA instead of other oxidation productions. Modelling does show some clue, however without validation of the modelling results, it is less useful. Would the authors show some experimental evidence, eg. by ATR-FTIR, that middle OH instead of terminal one in glycerol is preferably adsorbed?

2) ESR results can show byproducts, not the intermediates which always have a quite short lifetime. The other control experiment using H₂O₂ as a feedstock is different from the real condition in which OH radicals are in-situ generated as the latter is a heterogeneous process instead of a homogeneous process. One method to prove the pathway is to add the OH radical scavenger, which might disclose the key species.

3) The last one is about the calculation of IPCE. Would authors show the detailed step of the calculation, including the light power at 420nm, DHA yield at this special wavelength etc?

Point-by-point responses to the reviewer's comments

We sincerely thank the reviewer for his/her careful review and valuable comments, which certainly help to improve the quality of our manuscript. Our point-by-point responses are presented below and all changes made in the revised manuscript have been highlighted in yellow.

Reviewer #1 (Remarks to the Author):

The referee has a few questions, which have not been properly addressed in the revised.

1) These IR spectra, isotope-labeling experiments and ESR measurements do shine light on the possible reaction pathway, while cannot disclose the key information related to the selectivity to DHA instead of other oxidation productions. Modeling does show some clue, however without validation of the modeling results, it is less useful. Would the authors show some experimental evidence, eg. by ATR-FTIR, that middle OH instead of terminal one in glycerol is preferably adsorbed?

Response: We appreciate the questions raised by the reviewer.

We studied the ATR-FTIR spectra of BiVO₄ photoanode after treating it with glycerol solution at different pH. As shown in Fig. S7d and Fig. R1, the peaks at 994 cm⁻¹ and 1043 cm⁻¹ are attributed to the vibration modes of C-O bond in primary alcoholic group, and the peaks at 1110 ~ 1120 cm⁻¹ are attributed to that of the secondary alcoholic group.^{R1} When pH decreases, the C-O bond vibration peaks of glycerol increase distinctly. At the same time, the C-O bond vibration peak of the secondary alcoholic group shifts from 1109 cm⁻¹ to 1118 cm⁻¹ when pH decreases to 2, while the C-O bond vibration peaks of primary alcoholic group just shift by about 1 cm⁻¹. It indicates that C-O bond vibration frequency of the secondary alcoholic group is enhanced distinctly when pH decreases, which can be attributed to the analogous inductive effect by the enhanced electrostatic attraction between middle hydroxyl group and BiVO₄.^{R1} In contrast, the C-O bond vibration peaks of primary alcoholic group are only affected slightly due to the relatively poor adsorption of the terminal hydroxyl group on BiVO₄. The photoelectrochemical oxidation of glycerol-to-DHA conversion selectivity is 63.6% at pH = 2 and 1.2 V vs. RHE, while the middle hydroxyl group only accounts for 1/3 of all hydroxyl groups in glycerol. In combination with DFT results, we can conclude that middle hydroxyl group would preferably adsorb on BiVO₄ stronger as compared with terminal hydroxyl group.

Figure R1. ATR-FTIR absorbance spectra of BiVO₄ photoanode treated with glycerol solution at pH = 2, 5 and 7. Characteristic IR absorbance peaks of glycerol, Na₂SO₄ and H₂SO₄ are labeled with solid line, dotted line and dashed line, respectively.

2) ESR results can show byproducts, not the intermediates which always have a quite short lifetime. The other control experiment using H₂O₂ as a feedstock is different from the real condition in which OH radicals are in-situ generated as the latter is a heterogeneous process instead of a homogeneous process. One method to prove the pathway is to add the OH radical scavenger, which might disclose the key species.

Response: We appreciate the question raised by the reviewer.

We used *t*-butanol as the OH radical scavenger and studied the catalytic performance of BiVO₄ photocatalysts as shown in Supplementary Table S3 and Table R1. The results show that after adding *t*-butanol, the production of DHA was not affected too much, while after adding hole scavenger, ammonia formate, DHA could not be produced. It indicates that holes should directly participate in the glycerol selective oxidation reaction while hydroxyl radicals play less important role to selectively oxidize glycerol to DHA.

Table R1. Photocatalytic performance of glycerol oxidation by BiVO₄ photocatalysts in 0.5 M Na₂SO₄ at pH = 2 in a single cell under AM 1.5G, 100 mW/cm² illumination with hole scavenger or hydroxyl radical scavenger.

Production rate / mmol h ⁻¹ g ⁻¹	DHA	Formic acid	Glyceric acid	DHA selectivity / %
BiVO ₄ photocatalysts	0.138	0.144	0.072	39.0
+ 0.1 M ammonia formate	0	--	0.139	0
+ 0.1 M t -butanol	0.143	0.227	0.066	32.7

3) The last one is about the calculation of IPCE. Would authors show the detailed step of the calculation, including the light power at 420nm, DHA yield at this special wavelength etc?

Response: We appreciate the question raised by the reviewer.

As shown in the Supplementary Information, the incident-photon-to-current conversion efficiency (IPCE) was measured under monochromatic irradiation under one sun illumination (AM 1.5G, 100 mW/cm²) equipped with a monochromator at 1.23 V vs. RHE:

$$\text{IPCE} = \frac{\text{Number of all collected photogenerated charges}}{\text{Number of incident photons}} \times 100\%$$

$$= \frac{\left[\left(\frac{1240}{\lambda}\right) \times (J_{\text{light}} - J_{\text{dark}})\right]}{P} \times 100\%$$

where λ is the wavelength of monochromatic light, J_{light} is the photocurrent density under irradiation, J_{dark} is the current density under dark condition and P is the incident monochromatic light power density.

The light power of the light spot (0.2 cm²) by monochromator at 420 nm is 0.051 mW.

The results in Fig. 3c and d show that BiVO₄ photoanode has excellent photoresponse from 300 nm to 500 nm. In this range, BiVO₄ could provide photogenerated holes with sufficient energy to overcome the energy requirement for formation of radicals, which is ~1.5 eV. The oxidation of glycerol to DHA in this range is regarded nearly identical.

Faradaic efficiency was calculated by:

$$\begin{aligned} \text{Faradaic efficiency}(DHA) &= \frac{\text{Number of holes to oxidize glycerol to DHA}}{\text{Number of all collected photogenerated holes}} \times 100\% \\ &= \frac{e_{DHA} \times n_{DHA} \times N}{Q/n} \times 100\% = \frac{2 \times c_{DHA} \times V \times N}{Q/n} \times 100\% \end{aligned}$$

where e_{DHA} is the number of holes required to oxidize one glycerol molecule to DHA, N is Avogadro's constant, Q is the quantity of electric charge and n is the elementary charge.

Incident photon-to-DHA quantum efficiency (QE-DHA) was evaluated by multiplying IPCE with Faradaic efficiency of DHA:

$$\begin{aligned} \text{QE}(DHA) &= \frac{\text{Number of holes to oxidize glycerol to DHA}}{\text{Number of incident photons}} \times 100\% \\ &= \text{IPCE} \times \text{Faradaic efficiency}(DHA) \times 100\% \end{aligned}$$

References

- R1. Larkin, P. *Infrared and Raman spectroscopy : principles and spectral interpretation*. Elsevier (2011).

REVIEWERS' COMMENTS:

Reviewer #1 (Remarks to the Author):

The referee think the quality of the work has been substantially improved after a few cycles of revision although there are some minor points which were not fully addressed. However it does not affect the excellent quality of the paper and thus would suggest to accept it.

Point-by-point responses to the reviewer's comments

Reviewer #1 (Remarks to the Author):

The referee think the quality of the work has been substantially improved after a few cycles of revision although there are some minor points which were not fully addressed. However it does not affect the excellent quality of the paper and thus would suggest to accept it.

Response: We greatly appreciate the reviewer's careful review and valuable comments, which certainly help to improve the quality of our manuscript. We warmly thank the reviewer for recommending our work for publication in Nature Communications.